# Human Health Risk Assessment for Exposure to Heavy Metals via Dietary Intake of Rainbow Trout in the Influence Area of a Smelting Facility Located in Peru

**DOI:** 10.3390/toxics11090764

**Published:** 2023-09-08

**Authors:** Richard Peñaloza, María Custodio, Carlos Cacciuttolo, Fernán Chanamé, Deyvis Cano, Fernando Solorzano

**Affiliations:** 1Environmental Science & Health—ESH Research Group, Facultad de Medicina Humana, Universidad Nacional del Centro del Perú, Av. Mariscal Castilla N° 3909, Huancayo 12006, Peru; mcustodio@uncp.edu.pe (M.C.); e_2021101590k@uncp.edu.pe (F.S.); 2Civil Works and Geology Department, Catholic University of Temuco, Temuco 4780000, Chile; 3Facultad de Zootecnia, Universidad Nacional del Centro del Perú, Av. Mariscal Castilla N° 3909, Huancayo 12006, Peru; fchaname@uncp.edu.pe; 4Programa Académico de Ingeniería Ambiental, Universidad de Huánuco, Huánuco 10001, Peru; deyvis.cano@udh.edu.pe

**Keywords:** health risk, smelting facility, bioaccumulation, heavy metal, aquatic ecosystem, rainbow trout

## Abstract

Abandoned mining–metallurgical sites can significantly impact the environment and human health by accumulating heavy metals in aquatic ecosystems. The water in the sub-basin near an abandoned smelting facility in the city of La Oroya, Peru, is primarily used for pisciculture. The objective of this study was to assess the risk to human health from exposure to heavy metals via dietary intake of rainbow trout (*Oncorhynchus mykiss*) in the influence area of a smelting facility located in the central Andean region of Peru. The bioconcentration factor, biosediment accumulation factor, and consumption risks were evaluated using the Monte Carlo method. The results showed that the concentrations of elements (Zn > Pb > Cu > As) in rainbow trout muscle did not exceed the maximum limit (ML). However, the water significantly exceeded the ML for Pb in all sectors and As in the lower and middle parts of the river. The concentration of Pb in sediments also significantly exceeded the ML in the upper and lower parts of the river. Consequently, rainbow trout consumption in the study area presents risks to human health due to the bioaccumulation of heavy metals, with a 1.27% carcinogenic risk in samples from the lower part of the river.

## 1. Introduction

Contamination of aquatic ecosystems by heavy metals due to the accumulation of toxic substances in aquatic biota is a global problem of great interest [1,2,3]. This is due to their persistence, bioaccumulative capacity, and environmental toxicity [4,5,6], which can result from natural processes or anthropogenic activities [7,8]. These include improper management of industrial, agricultural, mining, or metallurgical wastes [9,10,11]. Among these factors, the latter two are the main contributors of heavy metals to water resources, leading to the exceedance of permissible heavy metal maximum limits set by local regulations, which can pose a significant threat to water quality and biodiversity [12].

While some metals (Mg, Fe, Zn, Cu, Se, Co, Mo, Cr, and Ni) are essential for normal biological functions in humans and animals [2,13,14], others, such as Al, As, V, Hg, Li, and Ag, lack any known role in biological functions, and can even be toxic in small quantities [13,15]. Metals like Cd and Pb are known to be teratogenic and can bioaccumulate and biomagnify in fish and other aquatic organisms, which can be transferred to humans through the food chain [16,17,18]. Consequently, the measurement and understanding of heavy metal concentrations in any aquatic ecosystem exposed to contaminants are highly relevant [18,19].

Fish are among the most used bioindicators when assessing water pollution [20]. This is not only due to their significance in monitoring potentially toxic elements [10,21], but also because of the health risk posed by exposure to harmful substances through the consumption of wild and farmed fish. Mercury (Hg), for example, can bioaccumulate in the muscle tissue of teleosts, presenting potential health risks to consumers upon ingestion [22]. Evaluations of heavy metal concentrations in wild and farmed fish reveal not only negative environmental pollution impacts but also adverse effects on fish and consumers when Hg concentrations exceed the environmental quality standard (EQS = 0.020 mg/kg) established by the European Union [23].

This highlights the importance of fish farming, which has been growing at an average annual rate of 7.5% over the last few decades [24], to protect food security, especially considering the declining numbers of wild fish due to overexploitation and pollution [25]. The high demand for fish as a food source, which reached 156 million tons in 2018 [24], coupled with the high consumption of wild-caught fish from unevaluated sources [26], underscores the need for speciation studies that serve as warning signals for both aquatic biota and public health when considering human consumption [23].

In the central Andes of Peru, the health status of many aquatic ecosystems remains understudied. However, water usage has increased, and untreated wastewater discharges have increased, leading to water quality alterations. In the sub-basins of rivers adjacent to an abandoned smelting facility in the city of La Oroya, water resources are utilized for various economic activities, including agriculture, mining, metallurgy, and pisciculture. These practices represent relevant risk factors for the examination and analysis of the biota inhabiting the area. Therefore, this study aimed to assess the risk of heavy metals in rainbow trout in the surrounding influence area of a smelting facility located in the central Andean region of Peru to human health following consumption. 

## 2. Materials and Methods

### 2.1. Study Zone

The study was conducted in the fish farms of the Tishgo River, which are located near the discontinued smelting facility in La Oroya city, situated in the central region of the Peruvian Andes, approximately 10 kilometers away. This river is one of the tributaries of the Mantaro River and is situated at coordinates 402,765 E and 8,738,887 S, zone 18 L, WGS84 (Figure 1). The area features rugged topography with slopes ranging from 2 to 10% [27], at an approximate altitude of 3750 m above sea level. The average minimum and maximum precipitation during the year are 6 mm and 115 mm in July and December, respectively. The average annual minimum and maximum temperatures are −4 °C and 17 °C in July and December, respectively.

### 2.2. Chemical Reagents and Chemical Grade

In this section, we provide a comprehensive list of the chemical reagents used in our study along with their corresponding chemical grades, ensuring the transparency and reproducibility of our methods:Nitric acid (HNO_3_): Commercial-grade nitric acid with a concentration of 65%.Hydrochloric acid (HCl): Analytical grade with a purity exceeding 99.5%.Sulfuric acid (H2SO_4_): Analytical grade with a purity between 95 and 97%.

All chemical reagents were handled and disposed of strictly with established laboratory safety protocols. The selection of analytical-grade chemicals ensured the integrity of our procedures and the accuracy of our measurements.

### 2.3. Sampling and Analytical Determination

The evaluated areas were divided into three sectors: upper, middle, and lower, as shown in Figure 1. A total of 12 water and sediment samples were collected from each sector. Water samples were collected using pre-treated 1.5 L plastic bottles, rinsed with distilled water. Sediment samples were collected using a modified Ekman grab at the same locations as the water samples in each sector. Additionally, 12 fish samples were collected from each sector using fishing nets and traps. All water, sediment, and fish samples were transported under refrigeration conditions to the laboratory on the same sampling day.

#### 2.3.1. Analysis of Heavy Metals and Arsenic in Water

Sample preparation involved taking 250 mL of water in a beaker and boiling it down to 100 mL. Subsequently, 5 mL of ultrapure nitric acid and 5 mL of hydrochloric acid were added to achieve complete oxidation and reduce interference caused by organic matter. The preparation was boiled, allowed to cool, and then 10 mL of distilled water was added. It was then filtered and measured in a 100 mL flask with 1% nitric acid. The quantification of arsenic (As), lead (Pb), and zinc (Zn) concentrations was performed using an atomic absorption spectrophotometer with a flame. Specific wavelengths were selected for each element being analyzed, and the spectrophotometer was calibrated using standard solutions of known concentrations of arsenic, lead, and zinc.

#### 2.3.2. Analysis of Heavy Metals and Arsenic in Sediment

A 1.0 g sediment sample was taken and placed in 100 mL beakers. To eliminate the present organic matter, 10 mL of nitric acid was added, allowing it to act on the sample. Then, 10 mL of hydrochloric acid was added, and it was allowed to interact with the sample for one minute. Subsequently, the containers with the samples were heated until a pasty consistency was obtained. Once this stage was reached, cooling was allowed, and an additional 10 mL of hydrochloric acid was added to ensure the complete dissolution of sample residues adhered to the glass walls. After completing the dissolution process, the samples were transferred to 100 mL flasks and properly homogenized. Distilled water was added to each flask to obtain a final volume of 100 mL. Subsequently, the samples were filtered to remove solid particles and obtain a clear liquid solution free from impurities.

For the quantification of Cu (copper), Pb, Zn, and As concentrations, the analysis utilized standard solutions that were prepared beforehand. These standard solutions contained known concentrations of each individual element, 10%, 20%, 30%, and 40%. Absorbance measurements were taken for each sediment sample and the standard solutions at the same predetermined wavelengths. By comparing the absorbance readings of the samples with those of the standards, the concentrations of Cu, Pb, Zn, and As in the analyzed sediments were determined.

#### 2.3.3. Analysis of Heavy Metals and Arsenic in Fish

The muscle tissue was extracted from the dorsal region of rainbow trout using a stainless-steel knife. Subsequently, the samples were placed in clean, high-density polyethylene containers for preservation. To maintain the integrity of the samples, they were stored at a temperature of −20 °C until the time of analysis. The preparation of muscle samples for residue analysis involved an acid digestion process. A 0.5 g sample of muscle tissue was taken and placed in a digestion flask. Then, 5.0 mL of concentrated ultrapure nitric acid (33%) and 5.0 mL of concentrated sulfuric acid were added to initiate the digestion reaction.

Once the reaction was initiated, the digestion flask was heated and cooled to allow proper digestion of the samples. Additional concentrated nitric acid was added, and the flask was heated at 120 °C for 6 h to complete the digestion process. Afterward, the samples were quantified to determine the concentrations of the target elements in the muscle of the rainbow trout. Following the digestion process, a portion of the digested sample was transferred to a 50.0 mL volumetric flask. The sample was brought to volume by adding approximately 35 mL of distilled water to each 0.5 g sample of rainbow trout muscle tissue. Once the sample was properly prepared, quantification was performed using atomic absorption spectrophotometry to determine the concentrations of target elements in the muscle of the rainbow trout.

### 2.4. Bioconcentration Factor (BCF)

The bioconcentration factor (*BCF*) is characterized as the absorption of contaminants from the dissolved phase [28,29] and can be calculated using the following equation:(1)$$BCF=\frac{C}{C_{w}}$$
where *C* represents the concentrations of contaminants in organisms (mg/kg) at equilibrium, and $C_{w}$ is the concentration of contaminants in the water (mg/L).

### 2.5. Biosediment Accumulation Factor (BSAF)

The biosediment accumulation factor is the potential of organisms to concentrate heavy metals in their bodies from sediment [30]. Therefore, it provides an indication of the rate of absorption and excretion of a substance by a living organism [31]. This was calculated using the equation:(2)$$BSAF=\frac{C_{MF}}{C_{MS}}$$
where $C_{MF}$ represents the concentration of the heavy metal in the organism and $C_{MS}$ the concentration of heavy metal in the sediment.

### 2.6. Risk Assessment for Consumption of Muscle of Rainbow Trout

The non-carcinogenic risk, Estimated Weekly Intake (*EWI*), Total Hazard Quotient (*THQ*), and Hazard Index (*HI*) values were calculated according to the equations described by Simukoko [23]. The Target Hazard Quotient is the ratio between the Estimated Daily Intake (*EDI*, mg/kg/day) of the metals, defined as the absorption dose multiplied by the absorption efficiency (Arm) of the metals in the human gastrointestinal tract [32], and the Oral Reference Dose (*RfD*, mg/kg body weight/day). The values of *EWI* and *EDI* (Estimated Daily Intake) for each element were compared with the Provisional Tolerable Weekly Intake (PTWI) and PTDI (Permissible Tolerable Daily Intake) established by the World Health Organization (WHO) [33]. 

The Target Hazard Quotient is the ratio between the Estimated Daily Intake (*EDI*) and the Oral Reference Dose (*RfD*, mg/kg body weight/day) [21,23], where the *RfD* is the level or dose of daily exposure (usually expressed in milligrams of toxic chemical substance per kilogram of body weight per day) for the human population [34]. Finally, the Hazard Index (*HI*), also referred to as the Total Hazard Quotient (*THQ*), is derived from the summation of individual *THQ* values corresponding to the metals. *THQ* and *HI* values > 1 indicate a risk of developing non-carcinogenic effects over a lifetime [35].
(3)$$EDI=\frac{MC\times IR\times EF\times ED}{BW\times AT}\times ADAF\times {AR}_{m}$$
where *MC* = mean metal concentration, *IR* = acceptable ingestion rate (0.34 kg/person/day), *EF* = exposure frequency (365 days/year), *ED* = exposure duration (74.8 years, which is the expected average lifetime), *BW* = average body weight (60 kg for an adult), *AT* = average exposure time for non-carcinogenic element (*EF* × *ED*), and *ADAF* = age-dependent adjustment factor (adult:1) [36]. The value of ${AR}_{m}$ for Pb is 33% and for As is 75% [37].
(4)$$THQ=\frac{E_{F}\times E_{D}\times F_{IR}\times C}{RfD\times W_{AB}\times T_{A}}\times 10^{-3}$$
where $E_{F}$ = exposure frequency, $E_{D}$ = exposure duration, $F_{IR}$ = fish ingestion rate, *C* = mean metal concentration, *RfD* = oral reference dose, *W_AB_* = average body weight of an adult, and *T_A_* = average exposure time with non-carcinogenic effect $\left(E_{F}*E_{D}\right)$
(5)$$HI=\sum_{i=1}^{n}{THQ}_{i}$$

The carcinogenic risk (*CR*) was evaluated to assess the possibility of cancer occurrence in individuals over their lifetime due to exposure to carcinogenic agents. The acceptable range of carcinogenic risk is from $10^{-4}$ to $10^{-6}$, and *CR* values higher than $10^{-4}$ are likely to increase the probability of carcinogenic hazard impact [38]. The *CR* of arsenic and lead was calculated using Equation (6):(6)$$CR=\frac{E_{F}\times E_{D}\times CSF\times EDI}{T_{A}}\times 10^{-3}$$
where *CSF* = cancer slope factor of cancer-causing agents (mg/kg-day)^−1^ which, for this study, was available for Pb (0.0085 mg/kg-day)^−1^ [39] and As (1.5 mg/kg-day)^−1^ [39,40] according to the database of the Integrated Risk Information System USEPA; *EDI* = estimated daily intake (EWI/7), while the total cancer risk (*TCR*) due to the consumption of rainbow trout from the evaluated fish farms was calculated as the sum of individual cancer risks [23], using Equation (7).
(7)$$TCR=\sum_{i=1}^{n}{CR}_{i}$$

### 2.7. Statistical Analysis

The data were analyzed using R software, R version 4.3.1 and R version 4.2.3 (R Core Team, Vienna, Austria) [41], which allowed us to perform the analysis with non-parametric Kruskal–Wallis tests [42] since the data did not have a normal distribution for the differences between the sampling sectors for each evaluated element. Additionally, medians were compared with maximum limits for water, sediment, and rainbow trout muscle using the Wilcoxon test [43]. Furthermore, the Spearman correlation test was used to establish relationships between heavy metals in water, sediments, and fish [31].

This study employed the Monte Carlo method for risk analysis through consumption with a 95% confidence level and a total of 10,000 simulations. The Monte Carlo method is a widely used technique in risk analysis in various fields, including risk assessment in projects [44]. It involves generating multiple random scenarios based on probability distributions and calculating the results for each scenario. This allows one to obtain an estimate of the probability of event occurrence and evaluating the impact of different variables on the outcome. Risk Simulator 2020 V 16 software (Microsoft Office, Washington, DC, USA) [45] was used for this purpose.

## 3. Results

### 3.1. Concentration of Heavy Metals and Arsenic

The medians of the concentration of heavy metals in water, sediment, and muscle of rainbow trout are provided in Table 1. The concentrations in rainbow trout muscle were, in descending order: Zn > Pb > Cu > As. The highest concentrations of Cu, Pb, and As were measured in the lower course, representing the highest-risk zone with 0.21, 0.24, and 0.11 mg/Kg wet weight (*w*/*w*), respectively. In the middle sector of the river course, Cu concentrations closely resembled those measured in the lower sector, at 0.20 mg/Kg, while Zn concentrations were 2.57 mg/Kg. However, there was a significant disparity in As concentrations. The middle sector and upper course had values of 0.02 mg/Kg and 0.01 mg/Kg, respectively.

Concerning the element concentrations in water, they were measured in the following descending order: Zn > Pb > As > Cu. The concentrations of Cu and Pb exhibited similar values, both having medians of 0.02 and 0.03 mg/L, respectively. In the case of As, the highest value was documented in the lower course at 0.03 mg/L, while the upper course displayed a concentration of 0.01 mg/L, and the middle course recorded 0.02 mg/L for this element. However, the concentration of Zn in the water of the lower course tended to be the lowest, with 0.05 mg/L. 

The concentrations of heavy metals in the river sediments were measured in descending order: Zn > As > Pb > Cu. Among the different sections of the river, the lower course exhibited the highest concentrations for Cu, Pb, and Zn, with mean values of 22.24, 47.69, and 231.04 mg/Kg, respectively. Significant differences were observed between this lower course and the other sectors of the river, particularly for Pb and Zn concentrations.

### 3.2. Bioconcentration Factor of Heavy Metals and Arsenic in Muscle

Kruskal–Wallis analysis indicated that there were no significant differences (*p* > 0.05) in the bioconcentration factor (BCF) values among the sampling points for each evaluated element (Figure 2; Appendix A). The highest values in the distribution of BCF in the upper, middle, and lower courses were observed for Zn (with medians of 40.57%, 37.90%, and 40.87%, respectively), followed by Cu (with medians of 10.77%, 9.97%, and 7.55%, respectively), Pb (with medians of 6.42%, 6.67%, and 9.35%, respectively), and As (with medians of 1.12%, 1.44%, and 3.29%, respectively).

### 3.3. Biosediment Accumulation Factor (BSAF) of Heavy Metals and Arsenic in Rainbow Trout Muscle

Unlike the bioconcentration factor of heavy metals, the distribution of BSAF values significantly differed between rainbow trout muscle (*p* < 0.05) along the river course for each evaluated element (Figure 3; Appendix A). For Cu, the highest BSAF was measured in the middle course of the river, with a median of 0.0127%. The upper and lower courses of the river obtained similar values (*p* > 0.05), with 0.0097% and 0.0109%, respectively. For Pb, equivalent values were measured in all sampling sectors (*p* > 0.05), with medians of 0.005%, 0.0064%, and 0.0045% in the upper, middle, and lower courses, respectively. The lower course exhibited notably lower values of Zn, with a median of 0.014%, which significantly differed from the other sectors (*p* < 0.05). Concerning As, BSAF showed a marked increase (*p* < 0.05) in the lower course of the river, displaying a median of 0.0026%. This was in contrast to the trend observed in the other sectors.

### 3.4. Relationship between Heavy Metals in Water, Sediment, and Muscle

According to the Spearman correlation analysis, As in rainbow trout muscle had a positive and significant correlation (*p* < 0.05) with Cu, Pb, Zn, and As in sediment (Figure 4). Additionally, As in muscle had a positive and significant correlation with Cu in water. Pb in muscle also exhibited a positive correlation (*p* < 0.05) with As in water and As in muscle. Furthermore, Zn in muscle showed a negative correlation (*p* < 0.05) with Zn and Pb in sediment.

Cu in water was positively correlated (*p* < 0.05) with As in water and negatively correlated (*p* < 0.05) with Zn in water too. As in water exhibited significant correlations with As, Zn, and Pb in sediment. Moreover, Cu in sediment showed a positive correlation (*p* < 0.05) with Zn and Pb in sediment, while Pb in sediment was positively correlated (*p* < 0.05) with Zn in sediment.

### 3.5. Risk Assessments on the Consumption of Rainbow Trout

Metal-specific Target Hazard Quotients (THQ) values quantify the potential health risks associated with each metal. Pb was the determining metal that contributed the most to the Hazard Index (HI), followed by As, Zn, and Cu (Table 2). This decreasing distribution, according to the contribution, was consistent in all evaluated sectors.

In the lower sector, there was a risk probability of 1.27% found for the exceedance of the established threshold. However, in the middle and upper sectors, there were no significant risks identified. These results highlight the importance of the Monte Carlo analysis as an effective tool for assessing health risks in cases of exposure to heavy metals (Figure 5).

## 4. Discussion

Rainbow trout muscle exceeded the maximum allowable limits for Pb and As of 0.015 mg/L and 0.01 mg/L, respectively, according to the quality standards set by the USEPA [48]. This excess was especially evident in the middle and lower sectors, which are tributaries to the Mantaro River, an aquatic ecosystem with mining contamination [50,51], which could have led to long-term accumulation. Subsequently, this contamination might lead to the gradual release of Pb and As. This accumulation of heavy metals in the environment may result from various activities associated with mining–metallurgical operations, such as atmospheric emissions, slag production, and disposal of mine waste, leading to long-term pollution of ecosystems. Improper management of mining–metallurgical waste can exacerbate the pollution problem [52].

Specifically, the concentrations of Pb were found to be 47.68 mg/Kg in the upper part and 36.62 mg/Kg in the lower part, both of which significantly exceeded (*p* < 0.05) the permissible maximum limit of 35 mg/Kg. Additionally, for Zn and As, the values measured in the river significantly exceeded the permissible maximums at all sampling points (*p* < 0.05), with maximum values found to be 231 and 42.44 mg/Kg, respectively, which can be attributed to metallurgical operations involving polymetallic concentrates since the smelting plant consisted of three production circuits for copper, lead, and zinc. Copper production began in 1992, lead in 1928, and zinc in 1952 [53,54], and these processes typically involved smelting and refining. This was especially true for copper, where mixtures of recycled material, fluxes, and concentrates were prepared, and refining involved roasting the mixtures to remove impurities such as arsenic, sulfur, antimony, and lead [55,56], which resulted in slag. Studies elsewhere report that arsenic concentrations were measured from 1070 ppm in roasted slag piles to 2960 ppm in mine tailings and 10,400 ppm in smelting residues. It was concluded that these contaminants increased according to the degree of processing [57], which contains traces of lead and arsenic [58]. These aspects emphasize the importance of proper mine waste management, as improper disposal of mine waste materials from these activities can result in the release of Pb into the environment, leading to its deposition in the sediment [59]. This was the case in La Oroya city, where a report prepared by the United States Government’s Agency for International Development, Peru Mission, revealed that the large slag hills in La Oroya city were evidence of an accumulation of Pb [60], which remains apparent to this day.

Furthermore, the presence of Pb in the sediment can be influenced by the physicochemical properties of the sediment itself. Sediments with high organic matter content and high metal-binding capacity, such as silt/clay and organic-rich sediments, may have a higher affinity for Pb and facilitate its accumulation [61]. The binding of Pb to organic matter and the formation of insoluble metal sulfides in the sediment can contribute to high concentrations of Pb in the sediment [62,63]. Additional deposition of atmospheric Pb from sources such as vehicle emissions and industrial emissions can also contribute to sediment contamination [64]. The presence of Pb in the sediment can indicate historical inputs from these sources, as sediments can act as sinks for Pb over time [65].

The binding of Pb to organic matter in the sediment can affect its mobility and bioavailability, potentially reducing its impact on the water column [61]. However, under certain conditions, such as anaerobic environments, mobilization of Pb from the sediment to the overlying water can occur [66]. This can result in the release of Pb into the water, potentially leading to increased Pb concentrations in the water and posing risks to aquatic organisms.

Metals tend to accumulate in aquatic organisms, and the concentrations of some metals can be magnified through the food chain. Humans can be exposed to metals through their diet, and over time, metals can accumulate at potentially toxic concentrations [67]. The results revealed that the investigated fish species accumulated metals in their muscle tissue, with the highest concentrations of As, followed by Hg, Pb, and Cd [68]. Based on the measured concentrations in rainbow trout muscle in the study area, the concentrations of Cu, Pb, Zn, and As do not exceed the maximum permissible limits of 30, 0.3, 30, and 2 mg/Kg *w*/*w* (*p* > 0.05), respectively, as set by the Codex Alimentarius [46] and Food Standards Australia and New Zealand [47]. However, it was shown that the maximum concentration of lead in the lower course of the river was 0.24 mg/kg wet weight, which is close to the hazard limit, similar to what has been measured in studies on salmon in the Baltic Sea [69].

The relationship between lead concentrations in the water and sediment can provide information about potential sources and transport mechanisms of lead in the study area. However, the absorption and accumulation of lead in fish can be influenced by various factors, such as species, feeding habits, and exposure routes [70], with the food pathway being more significant in some cases [71,72]. For factors such as transfer and bioaccumulation, elements like Zn and Cu had the highest proportions and interactions with rainbow trout in the study area. For Cu and Zn, these essential trace elements are necessary for various physiological processes in animals, including enzymatic function, growth, and immune system function [73]. However, excessive accumulation of copper and zinc in rainbow trout can have negative effects. High levels of copper in the meat can cause oxidative stress and tissue damage [74]. Similarly, excessive accumulation of zinc can disrupt the balance of other essential minerals and interfere with metabolic processes [75].

The correlations found between arsenic in muscle tissue and arsenic in water, as well as copper, lead, zinc, and arsenic in sediments, suggest possible chemical interactions and associations between these elements. Fish can absorb arsenic through water and food sources, and its accumulation in muscle tissue can be influenced by the bioavailability and absorption mechanisms of arsenic in the aquatic environment [35]. Similarly, the positive correlation between arsenic in muscle tissue and arsenic, copper, lead, and zinc in sediment indicates that the concentrations of these elements in sediment may play a role in the bioaccumulation of arsenic in fish [76,77].

For this study, the THQ results did not exceed the limit value. However, it was shown that the concentrations of Pb and As significantly influenced the HI (Table 2), with the rainbow trout from the lower sector of the river having a value of 0.23, the highest value in the river course samples. 

In the lower part of the river, the Monte Carlo simulation showed a 1.27% risk of cancer for humans from exposure to heavy metals via dietary intake of rainbow trout, with Pb and As being the risk coefficients that most contributed to the health risk. This finding is consistent with previous research highlighting the potential carcinogenic effects of heavy metals, including Pb and As [78]. These differences in human health risks may be due to the closer proximity of the sampling point to the Mantaro River, the urban area, and the mining wastes near the abandoned smelting plant at La Oroya city.

## 5. Conclusions

Zn emerged as an important factor in the bioconcentration factor (BCF) in rainbow trout muscle concerning concentrations in water, while Cu and Zn were the main contributors to the biosediment accumulation factor (BSAF). Furthermore, a positive correlation was observed between arsenic in rainbow trout muscle and arsenic in water/sediment, as well as lead in sediment.

The observed disparities in metal concentrations within rainbow trout muscle could potentially be attributed to physiological responses, particularly evident in the cases of Pb and As, allowing for the study and monitoring of heavy metal concentrations in aquatic ecosystems of watersheds near mining activities. In this way, it is possible to understand the effects and impacts on the trophic chain of ecosystems in the studied watersheds. Future studies are necessary to comprehend the degrees of exposure and transportation mechanisms of contaminants.

Our study included Monte Carlo risk probability analysis, finding a 1.27% risk of cancer in humans due to exposure to heavy metals via dietary intake of rainbow trout in the lower sector of the Tishgo River. This indicates the likelihood of exceeding a specific risk threshold, with Pb and As showing a stronger impact. This value suggests that these contaminants play a crucial role in contributing to the overall health risk for humans associated with rainbow trout consumption in the influence area of the La Oroya smelting facility. Including this indicator is considered a relevant contribution to addressing vulnerable areas with higher risk, which will facilitate decision-makers’ better understanding of the socio-environmental impacts generated by these abandoned mining industries. Future research could focus on refining cancer risk assessments by incorporating consumption patterns, socio-environmental factors, and contamination sources, which would provide a more comprehensive understanding and guide targeted interventions.

## Figures and Tables

**Figure 1 toxics-11-00764-f001:**
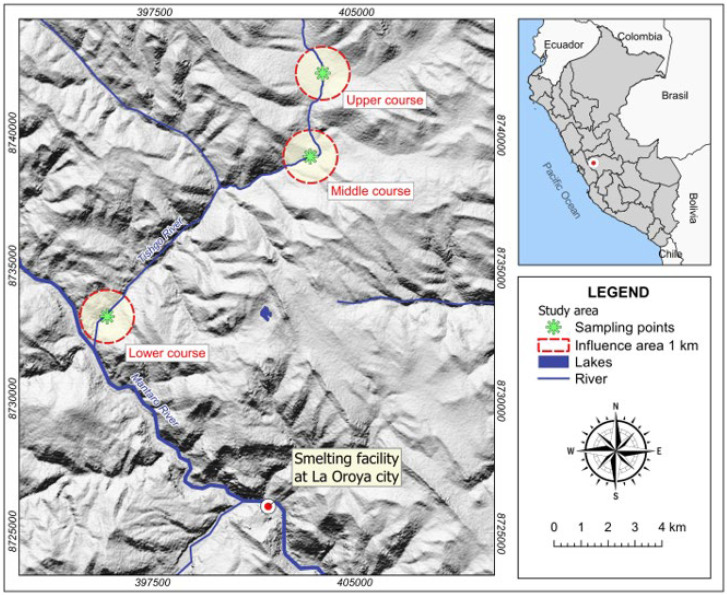
Study area and sampling sites, located near the city of La Oroya.

**Figure 2 toxics-11-00764-f002:**
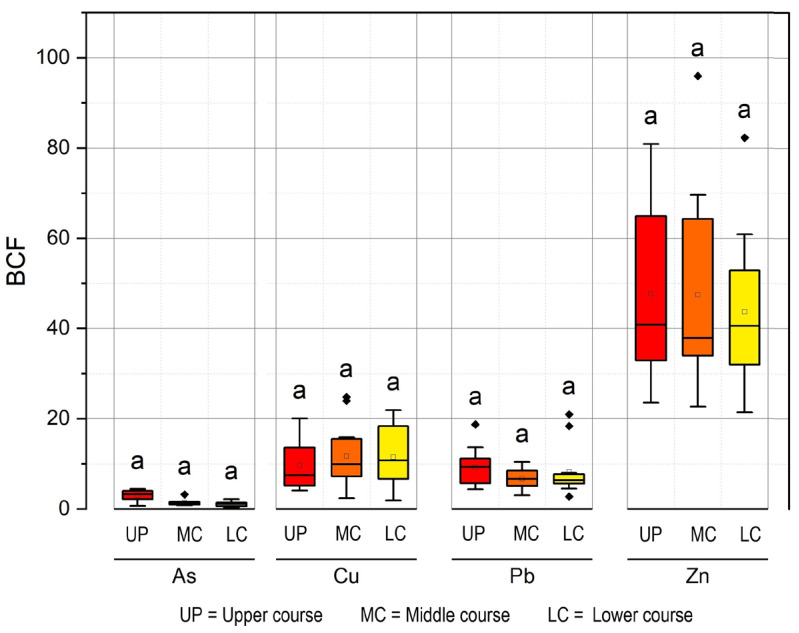
Distribution of bioconcentration factor of heavy metals and arsenic (letter “a” means that all sectors are significantly similar, *p* > 0.05).

**Figure 3 toxics-11-00764-f003:**
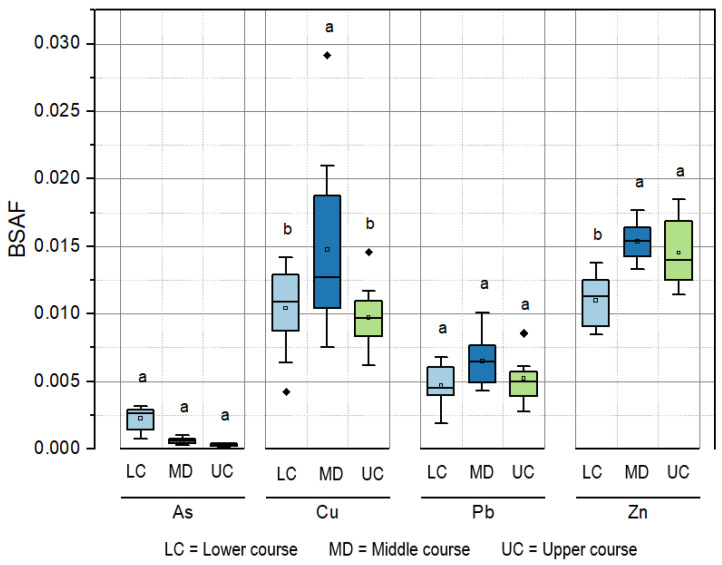
Distribution of biosediment accumulation factor (BSAF) of heavy metals and arsenic (letters “a” and “b” mean that all the sectors are significantly different, *p* < 0.05).

**Figure 4 toxics-11-00764-f004:**
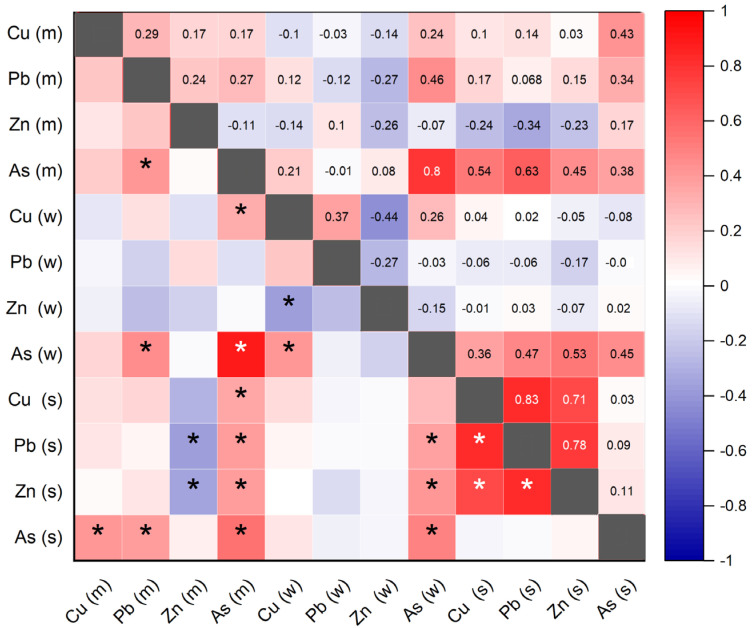
The Spearman correlation coefficient between heavy metals and arsenic of rainbow trout muscle (m), water (w), and sediments (s) (* significant correlation, *p* < 0.05; blue color gradient, negative correlation; and red color gradient, positive correlation according to Spearman).

**Figure 5 toxics-11-00764-f005:**
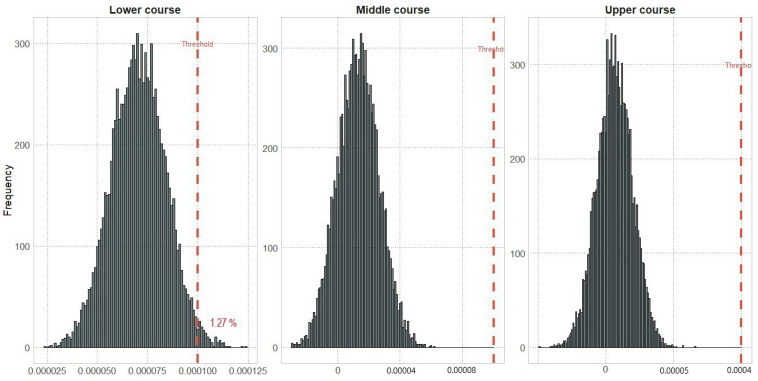
Monte Carlo simulation of the total cancer risk in humans by sector, following the exposure to heavy metals (As and Pb) via dietary intake of rainbow trout (lower course, middle course, upper course).

**Table 1 toxics-11-00764-t001:** Descriptive statistics of heavy metal concentrations in muscles of rainbow trout (mg/Kg *w*/*w*), water (mg/L), and sediment (mg/Kg) compared with maximum limits (ML) ^a–d^.

**Factor**	**Location**	**As**	Cu	Pb	Zn
Muscle	Lower course	0.11 ± 0.03	0.21 ± 0.11	0.24 ± 0.05	2.51 ± 0.08
	Middle course	0.02 ± 0.01	0.2 ± 0.11	0.19 ± 0.05	2.57 ± 0.09
	Upper course	0.01 ± 0.003	0.15 ± 0.15	0.17 ± 0.04	2.51 ± 0.07
	ML	${2\;}^{b}$	${30\;}^{a}$	${0.3\;}^{a}$	${30\;}^{a}$
Water	Lower course	0.03 * ± 0.005	0.02 ± 0.012	0.03 * ± 0.009	0.05 ± 0.031
	Middle course	0.02 * ± 0.003	0.02 ± 0.016	0.03 * ± 0.008	0.07 ± 0.025
	Upper course	0.01 ± 0.003	0.02 ± 0.017	0.03 * ± 0.01	0.06 ± 0.028
	ML	${0.01\;}^{c}$	${1.3\;}^{c}$	${0.015\;}^{c}$	${5\;}^{c}$
Sediment	Lower course	40.25 * ± 3.66	22.24 ± 2.32	47.68 * ± 9.97	231.04 * ± 39.08
	Middle course	42.44 * ± 3.41	15.22 ± 1.41	31.75 ± 2.92	169.83 * ± 14.54
	Upper course	32.53 * ± 2.74	18.39 ± 2.85	36.62 * ± 7.2	180.11 * ± 24.3
	ML—ISQGs	${5.9\;}^{d}$	${35.7\;}^{d}$	${35.0\;}^{d}$	${123\;}^{d}$

FAO/WHO ^a^ [46], FSANZ ^b^ [47], USEPA ^c^ [48], CEQGs ^d^ [49], * *p* < 0.05 significant difference with maximum limit (ML).

**Table 2 toxics-11-00764-t002:** Metal exposure assessment and hazard index by river sector.

Sector	THQ-Cu	THQ-Pb	THQ-Zn	THQ-As	HI
Lower course	0.0004	0.1954	0.0007	0.0299	0.2264
Middle course	0.0004	0.1547	0.0007	0.0054	0.1613
Upper course	0.0003	0.1384	0.0007	0.0027	0.1421

## Data Availability

The data associated with this research are available at Figshare under the following: https://doi.org/10.6084/m9.figshare.23735688, accessed on 26 September 2022.

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
