# Peer review of "Human Health Risk Assessment for Exposure to Heavy Metals via Dietary Intake of Rainbow Trout in the Influence Area of a Smelting Facility Located in Peru"

_toxics, 2023, doi:10.3390/toxics11090764_

Round 1

Reviewer 1 Report

The manuscript with ID (toxics-2550843) by Peñaloza and coauthors assessed the risk of consuming rainbow trout exposed to heavy Metals. The study is interesting and fits the scope of the journal. However, there are several points that the authors should revise before the manuscript is considered for publication. The authors should prepare a point-by-point to the comments raised by the anonymous reviewer.

General comment: -

§  Latin names should be written in italics throughout the whole manuscript.

§  Abbreviate the name of rainbow trout (Oncorhynchus mykiss) to (O. mykiss) after its first appearance in the text.

Title: -

§  It is too long a title. The title is three lines and a half. This is not interesting. The authors should write a concise and informative title.

§  I suggest using this title “Risk Assessment of Consuming Rainbow Trout Exposed to Heavy Metal Toxicity in Peru”.

Keywords: - The maximum number of keywords should be 5 or 6.

Results: -

§  I disagree with including references in the results section (Line 222 and Line 230). I understand that authors need to add reference levels to compare their findings. You can do this in the Discussion section.

Discussion: -

§  Line 327: The concentrations of Pb and Cu

§  Line 398: Delete (Fig. 5D).

References: -

§  The authors should use the reference style of MDPI. I suggest using EndNote or any reference management programs.

§  You should be consistent either to write abbreviated journal name or write the full name.

§  I do not know why the authors wrote the date and citation date in each reference. This is incorrect style.

Extensive editing of English language required

Reviewer 2 Report

This is a well-designed, interesting study about the distribution of heavy metals As, Cu, Pb, and Zn into water, sediment, and fish tissue in a stream (three locations) downstream abandoned mining wastes in central Peru. The methodology is clearly explained and follows standard procedures. The results are interesting, well presented and could apply to similar sites around the globe.

The figures (map and graphs) need some improvement and the text can also use some polishing, but nothing too major. The significant figures in Spearman coefficients need to be consistent, std deviation conventional expression is ±before the number, check the use of capitalized words and use terminology in a consistent manner. Units of measurement need to be changed to those required by the journal.

English is clearly and correctly written for the most part. The section of discussion includes repeated information and general information that belongs to the introduction. Also the section of Conclusions.

The accompanying marked file has some suggestions (take only those you want, you don’t have to accept all), starting with the title. For example, the study was conducted within a stream, and not in a watershed, therefore the term that better reflects the study is stream and not watershed in the title.

see above comments and suggested edits in the attached file

Round 2

Reviewer 1 Report

The authors properly addressed the comments raised by the anonymus reviewer.

Minor editing of English language required

Author Response

The english language in the article was revised by a native person from USA.